# Flowering and Fruiting of *Coffea arabica* L.: A Comprehensive Perspective from Phenology

**DOI:** 10.3390/plants14213396

**Published:** 2025-11-06

**Authors:** Carlos Andres Unigarro, Daniel Gerardo Cayón Salinas, Andrés Felipe León-Burgos, Claudia Patricia Flórez-Ramos

**Affiliations:** 1Discipline of Plant Physiology, National Coffee Research Center, Cenicafé, Manizales 170009, Colombia; 2Facultad de Ciencias Agrarias, Departamento de Ciencias Agrícolas, Universidad Nacional de Colombia, Palmira 763533, Colombia; dgcayons@unal.edu.co; 3Discipline of Crop Science, National Coffee Research Center, Cenicafé, Manizales 170009, Colombia; felipeleonb27@gmail.com; 4Plant Breeding, National Coffee Research Center, Cenicafé, Manizales 170009, Colombia; claudia.florez@cafedecolombia.com

**Keywords:** coffee, extended BBCH scale, growth and development, flower, fruit

## Abstract

In *Coffea arabica* L., the processes of flowering and fruiting unfold over time as a series of phenological events influenced by both biotic and abiotic factors. This sequence governs the plant’s reproductive cycle, directly affecting vegetative growth, crop productivity, and beverage quality. This review comprehensively addresses the developmental phases and provides descriptions of flower and fruit morphology, factors influencing flowering and fruiting, and competition for resource allocation, all of which are approached from a phenological perspective informed by the extended Biologische Bundesanstalt, Bundessortenamt and CHemical industry (BBCH) scale. The structural and emerging challenges that affect the sustainability of coffee cultivation should be effectively addressed to provide a foundation that supports the design of integrated strategies for the optimization of agronomic practices, increased yield, and genetic improvement.

## 1. Introduction

### 1.1. Origin and Distribution

Among the 124 species classified within the genus *Coffea* [1], only three are of commercial importance. Globally, the majority of commercial coffee bean production is derived from the species *Coffea arabica* L. (57%) and *Coffea canephora* Pierre ex A. Froehner (42%) [2]. In contrast, *Coffea liberica* accounts for less than 1% of commercial trade [3]. *C. arabica* is the only tetraploid species (2n = 4x = 44), having resulted from the natural hybridization of *C. canephora* and *Coffea eugenioides*, both of which are diploid (2n = 2x = 22) [4,5,6]. This hybridization is estimated to have occurred approximately 50,000 years ago [7,8] in the highlands (between 1600 and 2800 m) of the tropical rainforests of southeastern Ethiopia, the Boma Plateau in Sudan, and on Mount Marsabit in Kenya [4,9]. The wild coffee plant (*C. arabica*) is native to Ethiopia, where it was discovered approximately 850 AD and subsequently cultivated in the Arab colony of Harar. Although cultivation may have begun before 575 AD, the first written record dates back to Rhazes, a 10th century Arab physician [10]. The cultivation of *C. arabica* varieties commenced when wild coffee was introduced from Ethiopia to Yemen. There, the genetic base of *C. arabica* diverged into two primary botanical varieties, *C. arabica* var. Arabica (also known as *C. arabica* var. Typica Cramer) and *C. arabica* var. Bourbon (B. Rodr.) Choussy, from which most modern cultivars are derived [11].

The commercial production of *C. arabica* var. Typica outside of Yemen began in Sri Lanka in the 1660s and subsequently in Java in approximately 1700; a few plants were taken from Java to the Amsterdam Botanical Garden in 1706 [12]. Coffee was introduced to the Americas when seedlings were transported from Amsterdam to Suriname in 1718 and later disseminated throughout the Americas (Jamaica, Puerto Rico, Haiti, Cuba, Central America, Guyana, etc.) [11]. In 1727, coffee seeds were brought to the state of Pará in Northern Brazil, apparently from French Guiana [11]. The cultivation of *C. arabica* var. Bourbon originated from French plants brought from the coast of Mocha (Yemen) to the island of Réunion in 1715 and subsequently to Latin America and East Africa [12]. In Brazil, it was initially introduced between 1860 and 1870, from which it gradually expanded during the 19th century to other producing countries on the continent [13], including Colombia, where it was first introduced in Norte de Santander, and to the departments of Santander, Antioquia, Caldas, and Cundinamarca [14,15]. Currently, *C. arabica* L. is confined to the intertropical zone, which extends from latitude 20–25° N in Hawaii to 24° S in Brazil [16], with plantations in more than 45 countries worldwide, according to the ICO [2].

In Colombia, the varieties developed by the National Coffee Research Center (Cenicafé) have the Caturra variety—a spontaneous mutation first observed within a Bourbon population in the state of Minas Gerais, Brazil—and the Timor Hybrid (HdT) clone CIFC 1343, which originated from a natural cross between *C. canephora* and *C. arabica* in the 1940s, as their progenitors [13,17]. The resulting progenies from this cross were selected, and a set of them was used to develop the current composite coffee varieties (a mixture of different progenies) resistant to coffee leaf rust (*Hemileia vastatrix* Berk. & Broome; belonging to Basidiomycota, Pucciniales) [18]. In 2023, a survey in Colombia recorded 841,350 hectares of coffee cultivation located between latitudes 0°55′22′′ N and 11°16′37′′ N, accounting for the production of 14 million 60 kg bags of green coffee in 2024 [19].

### 1.2. Climatic Requirements for Cultivation

The optimal growth of *C. arabica* L. occurs at elevations between 1200 and 1950 m, with appropriate conditions found at approximately 1575 m [6,20], although the crop can be cultivated up to 2200 m [21]. In Colombia, Arabica coffee has been established at elevations ranging from 900 m to 2100 m [22]. *C. arabica* achieves moderate to very high production potential, with a sunshine duration of 1500 to 2100 h per year [23]. However, it can still grow with a minimum value of 915 h per year and a maximum value of 2330 h per year through the use of agronomic management strategies that facilitate adaptation in the coffee-growing region of Colombia [24]. Furthermore, Arabica coffee develops optimally under mean annual temperatures between 18 and 23 °C [25]; satisfactory yields have even been recorded at 24 to 25 °C in select cultivars under intensive management conditions [26,27]. Conversely, plant photosynthesis, growth, and productivity decrease sharply when the mean annual temperature falls below 17 °C [28]. Prolonged exposure to temperatures above 30 °C reduces growth and causes abnormalities, most notably leaf yellowing and the formation of tumors at the base of the stem [16,29]. The lower threshold temperature has been reported to be 10.0 °C (crop and flower to fruit) [30,31] or 10.5 °C (flower to fruit) [32], whereas the upper threshold temperature is 32.0 °C (crop) [30,31] or 32.4 °C (crop) [33]. In this context, the completion of the growth and development of the coffee crop from planting to the first harvest requires an accumulation of approximately 3250 GDDs (growth degree days), whereas approximately 2500 GDDs are necessary between the first flowering and harvest of *C. arabica* var. Caturra, assuming a base temperature of 10 °C [34].

The minimum rainfall required for the crop is 1250 mm year^−1^, with an optimum range of 1500 to 2000 mm year^−1^ [18,25]. In Colombia, where daily evaporation ranges from 3 to 4 mm, a dry spell of 30 to 40 days can negatively affect yield, particularly during the fruit-filling stage [24]. However, the daily evaporation also depends on other factors, such as the soil properties, cloudiness, atmospheric humidity, and the timeliness of agronomic management [35]. The vegetative growth of *C. arabica* requires atmospheric humidity below the saturation point [36], with the optimal range being 70 to 80% [37]. For *C. arabica* var. Catuaí cultivated under controlled conditions, mean wind speeds between 1 and 2 m s^−1^ favored dry mass accumulation and the net assimilation rate. In contrast, speeds greater than 2 m s^−1^ decreased the leaf area, internode distance, and length of plagiotropic branches [38]. In most of Colombia, the mean annual wind speed ranges from 1.4 to 2.8 m s^−1^ [24].

### 1.3. Phenology and Its Importance for Flowering and Fruiting

Phenology is the study of the timing of recurring natural phenomena in relation to climate [39,40,41]. It is influenced by abiotic factors, including the photoperiod, precipitation, temperature, and humidity, as well as by biotic factors such as genetics and intraspecific and interspecific competition for resources [40,42,43]. Given the diversity of terms used to describe specific plant structures in coffee, visual scales have been developed to classify and define the different phenological stages of the plants [44,45,46,47]. Among these scales, the extended Biologische Bundesanstalt, Bundessortenamt and CHemical industry (BBCH) scale for coffee is particularly useful, as it is an adaptation of a standardized system that is internationally recognized [44]. The extended BBCH scale is a system for uniformly coding phenologically similar growth stages across all monocotyledonous and dicotyledonous plant species, using clearly recognizable external characteristics [48], and it has been widely used in various studies [49]. The BBCH scale classifies plant phenology into a two-digit numerical code; the first digit (0–9) corresponds to the 10 principal growth stages, and the second digit (0–9) represents the secondary stages, which describe the short-duration developmental steps within each principal stage [48]. In this context, the phenology of flowering and fruiting plays a crucial role in determining the onset and duration of reproductive cycles in coffee [45,46].

Given its interdependence with abiotic and biotic factors, it directly impacts aspects such as vegetative growth, crop productivity, and beverage quality [50]. This information provides a theoretical foundation to detail the processes of flowering and fruiting. It enables the design of strategies to optimize the crop’s agronomic management or its response to edaphic and climatic conditions.

## 2. Flowering in Coffee

### 2.1. Development of Flowers

Flower formation is a complex process that encompasses two major physiological phases: flower initiation and flower bud development [51].

#### 2.1.1. Flower Initiation

The process begins with flower induction, which occurs in response to internal and external signals that regulate flowering time and determine the reproductive nature of the meristem [52]. Afterward, the process continues with anatomical changes—even in the absence of external stimuli—culminating in the formation of a primordium that is recognizable only under a microscope (initiation) [51,53,54,55]. The coffee plant is a short-day species; increases in photoperiod above 12 h of light delay flower initiation [56,57,58,59]. A photoperiod of less than 12 h was necessary for flower initiation [60]. Flower initiation does not occur when the photoperiod exceeds 13 h or when temperatures exceed 28 °C, regardless of the variety (e.g., Catuaí Rojo and Mundo Novo) [61].

#### 2.1.2. Flower Bud Development

The flower bud development phase can be divided into two subphases: the first involves the development of dormancy, while the second encompasses the breaking of dormancy and the resumption of growth until anthesis [16,62,63].

##### Development of Dormancy

The process begins with the identification of the primordium (determination) in the leaf axils of each node. This primordium is a 1 mm undifferentiated axillary bud (BBCH stage 51), characterized by a typical narrow, conical, green apex covered by bracts, which is observed as swelling in the leaf axils; at this point, the undifferentiated axillary bud is in a quiescent state and can potentially form lateral branches (vegetative growth) or flower buds (reproductive growth) or remain undifferentiated [44,51,61]. The development of BBCH stage 51 can last 30 ± 5 days [44]. The undifferentiated axillary buds grow in serial rows (“serial buds”) with varying degrees of development. Once a bud is defined as reproductive (initiation of differentiation, BBCH stage 53) (Figure 1), it is considered a flower bud and develops asynchronously from others [64,65]. The asynchrony among flower buds is considered the primary cause of nonuniform flowering in coffee in tropical regions [53]; however, the opening of inflorescences (BBCH stages 60–69) during a population-level flowering event is gregarious [66].

The flower bud passes through two distinct stages (BBCH-53 and BBCH-57) before achieving complete differentiation (BBCH stage 58). First, in BBCH stage 53, the flower bud emerges between the stipules, measuring 2–3 mm, as a dome covered by amber-colored mucilage secreted from the tip of the inflorescence; at this point, the bud is capable of producing flowers [44,61]. Second, in BBCH stage 57, the inflorescences emerge and become visible (7–9 mm) (Figure 1). Within these inflorescences, the green flower buds (3–6 mm) appear tightly clustered, with their corollas still adhering to one another and covered by mucilage and remaining in this stage for up to 45 days [44,61]. In BBCH stage 58 (Figure 1), inflorescences of 9–12 mm display separate flower buds (4–7 mm) as the mucilage disappears, along with their characteristic light green color, indicating the cessation of growth and entry into dormancy [44,61]. This stage is considered true dormancy, driven by high levels of endogenous abscisic acid (ABA) rather than by environmentally imposed quiescence [66]. At this stage, floral differentiation is complete [67]. Dormant flower buds (BBCH-58) are physiologically mature (Figure 1), which is why they are commonly referred to by terms such as “ready to flower,” “fully mature,” or “ripe to flower” [16,61,66]. Anatomical studies of the pedicels of dormant flower buds (BBCH stage 58) have shown that xylem connections are sparse and underdeveloped (with thick walls and small diameters), whereas phloem is abundant [62,68,69,70]. This limited xylem connectivity delays changes in the water potential of the buds, either maintaining them in a water deficit or limiting water loss to other parts of the plant (such as leaves) during periods of water stress [71]. Dormant flower buds (BBCH stage 58) cease their visible growth for a period that generally ranges from 30 to 120 days (Figure 1) [44,62]. This lack of growth occurs in response to the water stress necessary for normal flower development, as in its absence, flower buds may fail to develop or exhibit physiological disorders [63]. This topic will be discussed in greater detail later in this paper.

##### Breaking of Dormancy and Stimulus for Growth Resumption Until Anthesis

Following a dormant period of several weeks, the buds require a stimulus to resume growth. This stimulus is provided by the rehydration of tissues (e.g., rain, irrigation, fog, immersion of the cut ends of detached branches in water, or spraying the buds with water) and/or a sudden temperature drop of 4 °C per hour [62,68,72,73]. In the field, both signals typically occur together at the end of the dry season in what is known as “blossom showers” [16,62]. Usually, rainfall between 5 and 10 mm can trigger the resumption of growth [63,74], although greater amounts (>14 mm) may be required for adequate flowering [47]. Rehydration enables the direction of water movement to shift from net efflux from the buds to rapid influx into the buds, leading to the resumption of growth in the flower buds and their subsequent opening [75]. This process indirectly promotes changes in the hormonal balance between ABA and ethylene in the flower buds and leaves, although variations exist among *C. arabica* genotypes [76]. It has been demonstrated that following water deficit conditions, upon the onset of rehydration, ABA levels decrease in flower buds, leaves, and roots. This, in turn, increases ACC contents in flower buds and leaves, favoring greater expression of *CaACS1-like* and *CaACO1-like* and, consequently, leading to increased ethylene levels, although this is genotype-dependent [67,76]. Likewise, it was corroborated that ethylene sensitivity during the rehydration period is more closely associated with floral-opening responses in coffee plants. Therefore, this ethylene evolution in the aerial parts promotes foliar physiological processes, such as gas exchange. This facilitates rapid recovery of the plant’s water status following the water deficit, which, in turn, triggers the resumption of growth in the flower buds—a process that continues until anthesis [67]. From flower initiation to the end of dormancy, 90 to 150 days may elapse [65].

Once the flower buds resumption growth (48 h) [65], they rapidly gain fresh and dry mass over the next 6 to 12 days, depending on the temperature, after which their length increases three- to fourfold (to 6–10 mm) [44,62,68,73]. This increase in length occurs if the leaves underlying the flower buds are present during rehydration; otherwise, the buds may not respond easily to the water stimulus and may fail to resumption growth [75]. In the inflorescence, the flower buds acquire a whitish color and become fully developed, but their petals remain closed; this phenological stage is known as preanthesis (BBCH stage 59) (Figure 1) [44]. At this point, floral opening, or anthesis (BBCH stage 60) (Figure 1), is imminent and occurs within the next 3 to 4 days, when the petals, stamens, and pistil of a flower that is functional for fertilization become visible, as meiosis has been completed and the pollen grains have been released [44,63].

### 2.2. Flower Morphology and Characteristics

The coffee flower is classified as perfect, as it possesses both a pistil (stigma, style, and ovary) and stamens (filament and anther), and it is actinomorphic because of the symmetry of its petals. The flowers (4 to 5) are arranged in inflorescences, which originate from serial buds (3 to 5 buds) located in the leaf axils of the plant’s plagiotropic branches [44]. They can also be present on the stem (cauline flowers) [65,77]. Morphologically, the flower is composed of a lobate corolla, a calyx, five stamens, and a pistil [78]. The corolla (18.0–30.9 mm in diameter) begins as a 4 mm long green tube within the calyx; as it grows (7.0–7.2 mm), its tubular and cylindrical structure divides at the apex into five white lobes (petals) that are 8.0 to 14.5 mm long [8,65,79,80]. The stamens, which are 9.16 to 9.46 mm long, are inserted between the corolla petals, and each is composed of a short filament (4.1–5.0 mm) and a long anther (6.00–9.05 mm) that protrude from the corolla; the filament attaches to the midpoint of the anther. The anther, in turn, houses four pollen sacs [8,65,79,80,81] containing multiple pollen grains of up to six different forms (palynomorphs). Their equatorial (22.4 µm) and polar (21.5 µm) diameters are nearly circumferential [82]. The long (12.0–15.0 mm) and slender style is inserted above the ovary and below the corolla tube; at its upper end, it culminates in a bifurcated stigma (4.3–5.8 mm long), which protrudes slightly above the corolla tube [65,79,80]. The calyx (2 mm) is rudimentary, cup shaped, and located above the ovary, to which it is fused [79,81]. The ovary is positioned at the base of the corolla, and atop the pedicel, it is bilocular, with one ovule per locule; upon fertilization, these ovules develop into a globular drupe containing two seeds [65,83]. The pedicel (1–3 mm long), which supports the flower, is attached to the inflorescence by the calyculus [65,81].

The opening of the flower, or anthesis (BBCH stages 60–69), occurs in the early morning hours, releasing an intense, jasmine-like fragrance [78,81]. The viability of the pollen released from the anthers after anthesis is generally maintained for 24 to 36 h, while the stigmas remain receptive for 48 h—a period during which the pollen tubes can grow rapidly enough to reach the ovules [62]. The flower remains open for 2 or 3 days until the white corolla and stamens detach [65]. They initially remain attached to the style in what is known as the “fallen petals”, and subsequently, the style detaches and falls along with the corolla and stamens, leaving the ovary exposed [65]. Fertilization of the ovule generally occurs on the afternoon of the day of anthesis and is evidenced by a color change in the anthers from white to brown; conversely, if fertilization does not occur, the corolla, stamens, style, and stigma remain attached to the ovary [78]. In the fertilization process, self-pollination can contribute to 56.4% to 95.3% of the fruit set under Colombian conditions [84,85]. However, lower percentages (29.0–74.1%) [86,87] and high variability in wild *C. arabica* populations (3–91%) [88] have also been reported. Moreover, the percentage of fruit set attributable to insect-mediated cross-pollination can vary from 9.4% to 26.7% in Colombia, with these values being close to those reported in the literature (10–30%) [85].

### 2.3. Factors Conditioning Flowering

Coffee flowering is a complex process that remained poorly understood even at the beginning of the 21st century [36]. It is affected by factors such as the photoperiod, temperature, water availability, plant water status, and genotype [89].

#### 2.3.1. Photoperiod

Coffee has been classified as a “short-day plant” because flower initiation is stimulated by short days (long nights) and delayed by long days (short nights) [58]. Studies in this area have shown that water stress and temperature do not significantly affect the timing of flower initiation (BBCH stage 51), which only occurs in coffee when the photoperiod is less than 12 h [56,57,58,59,60]. In coffee, a correlation exists between latitude and flowering, driven primarily by the day length and soil moisture content. Thus, at latitudes below 2.0° N in Colombia, water deficit stress is more relevant than at locations above 4.5° N latitude, a decrease of up to 30 min in the photoperiod from June to December has a greater influence on the initiation of flowering [90]. The process of flower initiation (BBCH stage 51) begins four to five months before the main flowering events occur (January to March), which coincides with the decrease in day length that stimulates flowering at latitudes above 4.5° N [91]. It has recently been suggested that insolation (W m^−2^), rather than day length, is the factor responsible for flower initiation [92,93]. Daily insolation is a function of the day length and daily integrated irradiance (excluding atmospheric effects), which varies throughout the year and with latitude, depending on the angle at which the sun’s rays strike the Earth [94,95]. In this sense, changes in daily insolation trigger the molecular processes that enable reproductive development, making variations in light intensity due to cloud cover irrelevant [94,96]. The change in daily insolation is an indirect but significant factor involved in coffee flowering, as it acts by switching “on” or “off” the flower initiation phase (BBCH stage 51), and subsequent development leads to bud dormancy (BBCH stage 58), in turn dictating the expression or suppression of anthesis (BBCH stage 60), thereby affecting the frequency of cycles per year [93]. Furthermore, sunshine duration is positively correlated with the transition of buds from the dormant state (BBCH stage 58) to preanthesis (BBCH stage 59) [72], as well as with the total number of flowers [97].

#### 2.3.2. Plant Water Status and Water Availability

Multiple studies have indicated that a dry period of moderate to high intensity facilitates the conditioning of dormant buds (BBCH stage 58), making them more sensitive to breaking dormancy when this period is interrupted by rain or irrigation; this water stress also promotes greater synchrony of floral opening when more than 60% of the buds are in a dormant state [60,67,72,75,98,99,100,101,102,103,104,105]. Floral synchrony occurs because dormant buds (BBCH stage 58) do not advance to the next stage during dry periods, allowing buds at an earlier developmental stage to also reach dormancy and achieve more uniform flowering once the rehydrating effect of the first rains or irrigation occurs [101]. In contrast, when flower buds experience a dry period before reaching dormancy (BBCH stage 58), the stimulus from rehydration has no effect because the buds are still in a dormant state [106]. Therefore, the stimulating effect of rehydration is restricted to dormant buds (BBCH stage 58) [106] because these buds possess a well-defined vascular cylinder containing a developed secondary xylem through which water, nutrients, and the root-derived signaling metabolites responsible for resuming bud growth are transported [99]. Precipitation between 5 and 10 mm can provide sufficient rehydration to break bud dormancy (BBCH stage 58) and restart growth through anthesis (BBCH stages 60) [25,74].

With respect to the necessary stress, dormant buds (BBCH stage 58) must experience soil water potentials (Ψs) below −1.4 MPa and a predawn leaf water potential (Ψw) of −1.2 MPa before they can be stimulated by water application because at this threshold, water flow from the leaves to the buds tends to stop and reverse because the leaf water potential becomes equal to or more negative than that of the bud; this difference enables growth to resume in response to rehydration [75]. For example, a water deficit (Ψw > −2.0 MPa) in coffee plants affects ethylene levels in leaves and flower buds. This results from reduced expression of *CaACS1-like* and *CaACO1-like* genes involved in biosynthesis, as well as altered ethylene sensitivity due to changes in *caETR4-like* receptor genes. Consequently, flower buds remain dormant, yet this dormancy may support their competence to flower [67,107]. These responses have been linked when coffee plants undergo water deficit, there is a demonstrated relationship between more negative leaf water potentials (Ψw between −1.0 and −3.0 MPa) and increased ABA content in leaves and roots. This pathway inhibits *CaACS1-like* and *CaACO1-like* gene expression and raises levels of ACC (1-aminocyclopropane-1-carboxylic acid) and its conjugated forms, which may explain the reduced ethylene levels in flower buds [52]. Nevertheless, increased ACC concentrations in leaves and roots before flower bud rehydration are thought to act as biochemical signals that stimulate floral anthesis in coffee [76,108]. Although a Ψw threshold is considered a requirement, currently, a definitive consensus value has not been reached. However, floral opening is generally more common and synchronous at Ψw values below −1.0 MPa (e.g., Ψw = −1.15 MPa [109], Ψw = −1.2 MPa [75], Ψw = −1.7 MPa [105], Ψw = −2.0 MPa [110], Ψw = −2.3 MPa [67], Ψw = −2.5 MPa [60], Ψw = −2.65 [111]) than at higher Ψw values (e.g., Ψw = −0.71 [104], and Ψw = −0.8 MPa [99]). Similarly, anthesis (BBCH stages 61–69) can be characterized by mild, prolonged dry periods and intense, short-duration dry periods [99], as well as by endogenous plant factors [89]. On the other hand, the absence of a water deficit results in flower buds remaining dormant (BBCH stage 58) and failing to reach floral opening (BBCH stage 60) [91]. Eventually, their numbers begin to decline because of bud senescence [99].

From a large-scale perspective, the Intertropical Convergence Zone influences the intra-annual rainfall distribution in the Andean region of Colombia [24]. In turn, it affects the amount of rain during flowering periods and, consequently, the resulting floral expression (BBCH stages 60–69) [60,93]. In the coffee-growing zone of Colombia, the El Niño phenomenon—the warm phase of the El Niño-Southern Oscillation (ENSO) climate pattern—manifests primarily as reduced and less evenly distributed rainfall, which promotes the expression of anthesis (BBCH stages 60–69) in highly concentrated, large-magnitude flowering events [23]. This process occurs because a medium- to long-duration dry period allows more flower buds at different developmental stages to reach dormancy (BBCH stage 58), and subsequent rains then break this dormancy, leading to greater expression of flowering (BBCH stages 60–69) [23,52,60,112,113]. In contrast, the La Niña phenomenon, the cold phase of ENSO, reduces flowering [23]. Excessive rainfall during this phenomenon causes buds to remain dormant (BBCH stage 58) for longer periods, eventually decreasing their number because of senescence when they fail to break dormancy [99]. Furthermore, with an increase in the number of rainy days, flowering events become more dispersed, less concentrated, and of lower magnitude [24,114]. Similarly, prolonged water excess can promote the production of “star flowers” by inducing the premature resumption of growth in dormant buds (BBCH stage 58) before they have achieved the proper conditioning normally obtained during a defined dry period [79,115]. This situation is frequent during La Niña climatic events [115] or in areas of high precipitation where the incidence of this disorder can reach 12.9% to 17.8% of the total recorded flowers [79]. On the other hand, a water deficit can also affect the resumption of growth in dormant buds (BBCH stage 58) because of insufficient rehydration, resulting in “star flowers”. This phenomenon usually occurs when a low-intensity rainfall event occurs within a prolonged dry period accompanied by high temperatures [79].

#### 2.3.3. Environmental Temperature

Temperature is among the most critical factors influencing flowering phenological events [116,117]. In coffee, flower initiation occurred only when day/night temperatures were less than 33/28 °C [61]. In contrast, day/night temperatures of 23/18 °C were most suitable for the development of dormant buds (BBCH stage 58), resulting in fewer malformations, greater synchrony, and a greater number of inflorescences per node [61]. Similar results have been reported for dormant buds at day/night temperatures of 23/17 °C [118] and 27/17 °C [119]. Additionally, elevated day/night temperatures promote premature anthesis (BBCH stages 60–69). For example, at 30/24 °C, floral opening occurred eight days after the buds broke dormancy, whereas at 23/17 °C, it occurred after 11 days but without a reduction in the number of flowers [73]. Elevated temperatures during anthesis, especially under conditions of a prolonged water deficit, can also induce flower abortion [45,120]. A decrease in air temperature following a precipitation event plays an important role in breaking the dormancy of flower buds (BBCH stage 58) [73,106]. In Kenya, flowering has been observed when the air temperature decreases by more than 3 °C within a period of less than 45 min [121]. In Colombia, compared with higher temperatures, air temperatures less than 20 °C are associated with fewer flowering events [93]. Likewise, when the air temperature decreases during the night, the induced stomatal closure can produce stress that is potentially similar to that which dormant buds (BBCH stage 58) require for their conditioning (≈−0.8 MPa) [122]. Finally, coffee plants may exhibit greater sensitivity to thermal variation during short days than during long days [123]. The thermal time required from flower initiation until the bud enters dormancy (BBCH stage 58) is 840 GDD [30]. Dormant buds (BBCH stage 58) can remain in this state for 840 to 2562 GDD; then, once dormancy is broken by rainfall, anthesis (BBCH stage 60) occurs after an additional 100 GDD [30]. In Brazil, buds require an accumulation of 1579 GDD from flower initiation to reach dormancy (BBCH stage 58) and a minimum of 7 mm of rainfall to break it and achieve floral opening (BBCH stage 60) [124]. In both cases, the thermal time calculation assumed a base temperature of 10 °C. Quarterly accumulated thermal time has been shown to be directly related to the number of buds in preanthesis (BBCH stage 59) at some locations in Colombia [125]. The number of flowers in both agroforestry and full-sun systems decreases with increasing elevation, which is linked to a reduction in ambient temperature [126].

#### 2.3.4. Genotype

Highly contrasting flowering patterns have been reported among *C. arabica* L. genotypes at the same latitude. An example is the Semperflorens mutation, which is associated with continuous flowering events throughout the year in Brazil [120,127]. In contrast, commercial varieties such as Borboun typically flower two to three times per year under the same conditions [35,45,128]. However, among commercial *C. arabica* lines, flowering patterns do not differ significantly in terms of synchrony or temporal variability [93].

## 3. Fruiting in Coffee

### 3.1. Fruit Development and Growth

In *C. arabica*, the period from anthesis (BBCH stage 60) to fruit ripening (BBCH stage 88) lasts, on average, between 220 and 243 days after flowering (DAF) [11,36,62,129,130]. However, depending on the ambient temperature and humidity of the location (site), this duration can vary from 180 to 330 DAF [65,77]. Furthermore, fruit growth exhibits a double sigmoid curve pattern, which can vary because of environmental factors, the characteristics of the fruit and its parts, the sampling frequency, the statistical model used, and the genotype [36,130]. Regardless of variations in ripening time and growth patterns, fruit development can be divided into five phases.

#### 3.1.1. Pinhead Stage (56 DAF)

Once the ovaries are fertilized, they undergo rapid cell division [62], which culminates in the formation of fruits that are initially visible as small, yellowish berries (BBCH stage 70) [44]. In the following weeks, the mass or volume of fruits changes very little, reaching up to 10% of their final size (BBCH stage 71) (Figure 2) [44,131], primarily because of the development of the perisperm [132]. At this point, the fruits are referred to as “pinheads” [133]. In this phase, the fruit set is determined because the endosperm has been established. The growth rate has been reported to be low (1.8 mg day^−1^) [129], whereas the respiration rate is moderately high (1.1 mL O_2_ g^−1^ h^−1^) [134]. Furthermore, this stage has the highest stomatal density in the fruit during development, ranging from 140 to 160 stomata mm^−2^ [44,135].

#### 3.1.2. Rapid Swelling or Filling Stage (57 to 126 DAF)

Light green fruits grow rapidly in volume and fresh mass, a process driven by the predominance of cell expansion, until they reach approximately 30% of their final volume between 10 and 11 weeks after flowering (BBCH stage 73) (Figure 2) [44,129]. This growth is primarily due to the expansion of the pericarp [62]. At this stage, the endosperm becomes visually identifiable [132]. The fruit locules that will contain the beans subsequently swell due to the expansion of the perisperm (or integument) until the maximum bean size is reached [36]. At this point, the bean has a gelatinous and translucent consistency, constituting 50–60% of the final fruit volume (BBCH stage 75) [44,129]. The size to which the locules swell depends mainly on the plant’s water status, given that fruits accumulate 80% to 85% of their final water content during this phase [131]. The endocarp (parchment), which surrounds the locules like a line, begins to lignify [136]. The growth rate per fruit reported for this phase was 13.0 mg day^−1^ [129], while the respiration rate was 1.33 mL O_2_ g^−1^ h^−1^ at 30 °C [134].

#### 3.1.3. Suspended and Slow Growth Stage (126 to 140 DAF)

The growth rate of the fruit significantly decreases (0.3 mg day^−1^) [129] as the final fruit volume is reached [133]. However, its dry matter content remains low, representing 22% of the fruit’s fresh final mass [129]. The respiration rate decreases to 0.5 mL O_2_ g^−1^ h^−1^ at 30 °C [134].

#### 3.1.4. Endosperm Filling Stage (126 to 196 DAF)

In dark green fruits, the perisperm within the locules is gradually consumed and almost completely replaced by the endosperm; the remaining perisperm tissue forms a thin layer surrounding the endosperm, commonly known as the “silver skin” [132]. This process results in the formation of the beans that constitute the seed; they now have a solid consistency and constitute 70 to 80% of the final fruit volume (BBCH stage 77) [44,129]. During this period, a pronounced increase in dry mass occurs, reaching 36% of the fresh final mass of the fruit [129]. Dry mass is deposited mainly in the beans, which reach their final dry mass while the fruit is still green [62,133]. This phase culminates in the “physiological maturity” of the fruit, denoted by its olive-green coloration and a completely solid endosperm within the locules, and the fruit has reached 90% of its final volume (BBCH stage 79) (Figure 2) [44,129]. The endocarp surrounding the locules continues its process of lignification [11]. The growth rate per fruit was 0.4 mg day^−1^ [129], while the respiration rate was 0.3 mL O_2_ g^−1^ h^−1^ at 30 °C [134].

#### 3.1.5. Ripe Phase (196 to 224 DAF)

The pericarp (exocarp, mesocarp, and endocarp) exhibits changes such as the initial color transition from olive green to light red–yellow, along with increases in the fresh and dry masses, which result in a larger fruit volume (BBCH stage 81) [44,62,133]. Following this increase in mass, the intensity of the red or yellow color of the fruit deepens, and processes such as respiration and ethylene production increase notably; however, the fruit is not yet suitable for harvest (BBCH stage 85) [44,62,133,137]. Changes in color occur because of the degradation of chlorophyll, which occurs in parallel with the accumulation of anthocyanins during fruit ripening [138]. The fruit turns completely red or yellow, depending on the variety, and reaches its maximum mass (both fresh and dry); at this point, it is suitable for harvest (BBCH stage 88) (Figure 2) [44,129]. The respiration rate during this stage did not differ from that of the rapid endosperm filling phase [134].

### 3.2. Fruit Ripening Process

Fruit ripening is a coordinated, genetically programmed, and irreversible process involving a series of biochemical, physiological, structural, and organoleptic changes that lead to the development of a ripe fruit, which is attractive to consumers because of its desirable quality attributes [139,140]. Once the fruit has completely formed its endosperm (BBCH stage 79), it exhibits a notable increase in ethylene biosynthesis, which decreases as the fruit approaches maturity or the harvest point, at which point no loss in mass occurs and the ideal sensory characteristics are met (BBCH stage 88) [137,141]. This behavior, as well as the color change in the epidermis from olive green (BBCH stage 79) to the characteristic red hue of ripening (BBCH stage 88), has been associated with several factors in the coffee fruit: the regulation of genes involved in ethylene biosynthesis [137,142]; the peak respiration rate that precedes the increase in ethylene production [137,143]; and the positive effect of exogenous applications of “Ethephon” or “Ethrel” (2-chloroethyl phosphonic acid) [144,145,146]. Taken together, these characteristics suggest that coffee is a constitutively climacteric fruit [142,147]. However, “Ethephon” only accelerates the ripening of the pericarp without affecting the development of the bean (endosperm + embryo), which explains why the resulting beverage quality can be deficient [35].

Furthermore, not all *C. arabica* genotypes exhibit a standard climacteric response. Ripe fruits (BBCH stage 88) of the cultivar Catucaí 785-15 show a typical pattern for climacteric fruits, with notable increases in the rates of respiration and ethylene production, whereas the cultivar Acauã has lower levels of both [143]. In turn, these different levels of ethylene production may be directly associated with the ripening time of each cultivar. The higher ethylene production in Catucaí 785-15 fruits, which is associated with the expression of the *CaACO1-like* and *CaACO4-like* genes, may have promoted increased degradation of *CaETR4-like*, allowing for an increase in ethylene sensitivity and, consequently, early ripening; in contrast, in the fruits of Acauã, which failed to inactivate *CaETR4-like* expression, ripening was delayed [142]. Separately, sucrose accumulates in the pericarp and endosperm, reaching its maximum concentration when the fruit is ripe (BBCH stage 88) [132,148,149,150]. The sucrose content in ripe fruits (BBCH stage 88) depends mainly on the coffee species or variety [149] and reaches values between 7% and 10% (dry basis) in *C. arabica* [148,149,151]. The maximum sucrose concentration in coffee beans was reached more quickly in the early-ripening varieties Mundo Novo IAC 376-4 and Catuaí Vermelho IAC 144 than in the late-ripening variety Obatã IAC 1669-20 [152].

### 3.3. Fruit Anatomy and Morphology

The coffee fruit is a drupe that consists of a fleshy pericarp that surrounds a generally bilocular ovary, with each locule giving rise to one seed [11,135].

#### 3.3.1. Pericarp

The pericarp is composed of three tissue layers: the exocarp, mesocarp, and endocarp.

Exocarp or Epidermis: The primary role of this tissue is to provide mechanical protection for the inner layers of the fruit [153]. It is composed of a single, continuous layer of compact, isodiametric parenchyma cells that maintain a relatively constant size (15 µm, on average) during fruit development [135] and are protected by a waxy cuticle that changes during ripening [153]. This tissue grows via cell division, resulting in thicker outer cell walls and thinner inner walls. Chloroplasts are found in variable amounts within these inner regions [135,154]. From the pinhead stage (BBCH stage 71) to the endosperm filling stage (BBCH stage 79), large amounts of chlorophyll give the exocarp a green coloration, which progressively changes to a reddish hue upon reaching ripening (BBCH stage 88) [44,129,135], a shift caused by the accumulation of anthocyanins as chlorophylls are degraded [138]. In some natural mutations of the species, the fruit exocarp remains yellow at ripening (BBCH stage 88) because leucoanthocyanin is present instead of anthocyanin, allowing the yellow pigment luteolin to be exposed [11,44,132]. The exocarp possesses stomata similar to those of the leaves but in a lower quantity (approximately 5.5-fold fewer) (BBCH stage 88) [135]. The total number of stomata is established at the beginning of fruit formation and remains relatively constant throughout fruit ontogeny [155]. However, as the fruit expands in volume, the stomata move farther apart, causing the stomatal density to decrease. For example, in young fruits (BBCH stage 75), the stomatal density was 137 stomata mm^−2^, whereas in ripe fruits (BBCH stage 88), it was 44 stomata mm^−2^ [44,135]. In comparison, leaves exposed to full sun exhibit a stomatal density that varies between 200 and 260 stomata mm^−2^ [62,156].

Mesocarp: This tissue, located between the exocarp and endocarp, is composed of more than 20 layers of shape oval or round parenchyma cells, as well as vascular bundles that are immersed and arranged in a circular pattern [135]. During the rapid growth phase (BBCH stage 71, 60 DAF), the mesocarp cells rapidly divide. This division almost completely ceases by the end of the rapid growth phase (BBCH stage 75, 105 DAF), after which point the cells only expand in size [44,135]. The size of the parenchyma cells increases throughout fruit development, increasing from 32 µm at 60 DAF (BBCH stage 71) to 69 µm at 224 DAF (BBCH stage 88) [44,135]. The mesocarp contains chloroplasts until the end of the endosperm filling stage (BBCH stage 79, 195 DAF), at which point their numbers begin to decrease as the quantity of tannins increases [44,135]. From this time until the ripe phase (BBCH stage 88 and 240 days), the mesocarp rapidly increases in thickness because of hydration, which enlarges the cells closest to the endocarp [44,135]. The mesocarp is divided into two zones: an outer zone and an inner zone. The outer zone is composed of parenchymal cells with thick, compact cell walls until physiological maturity (BBCH stage 79), which gives the fruit a stiff consistency; subsequently, during ripening (BBCH stage 88), the action of pectinolytic enzymes thins these walls, causing the tissue to soften to an approximately 5 mm thickness [11,44,135,157]. It represents approximately 29% of the dry mass of ripe fruit (BBCH stage 88) [157] and is composed mainly of carbohydrates (44–50%), proteins (10–12%), fibers (18–21%), polyphenols (1.48%), and caffeine (1.3%) [158]. The inner zone, adjacent to the endocarp, is a mucilaginous tissue known as “mucilage” in ripe fruits (BBCH stage 88); it is formed by 2 or 3 layers of columnar, palisade-like tissue with a thickness between 0.5 and 2.0 mm, and it corresponds to 4% of the dry mass of the fruit [11,44,135,157]. The mucilage is composed of water (84.2%), protein (8.9%), sugar (4.1%), pectic substances (0.91%), and ash (0.7%) [159].

Endocarp: At the beginning of the rapid growth phase (BBCH stage 73), this tissue is composed of three to seven layers of irregularly shaped sclerenchymatous cells, which range in size from 7 to 15 µm and contain few chloroplasts; in subsequent phases (after 90 days, BBCH stages 73–88), the cells harden and transform into sclereids, forming the lignified seed coat known as the “parchment” [44,135]. The parchment layer is a thin, yellowish, brittle, paper-like polysaccharide with a thickness of approximately 100 to 150 µm; it represents 12% of the dry mass of the fruit and is composed mainly of α-cellulose (40–49%), hemicellulose (25–32%), lignin (33–35%), and ash (0.5–1.0%) [132,157,159,160,161]. The primary function of the endocarp is to protect the seed from mechanical damage and to serve as a barrier to the transfer of chemical compounds to and from the seed [11].

#### 3.3.2. Seed

After fertilization, while the first changes occur in the fertilized embryo sac, the surrounding perisperm undergoes a rapid increase in volume because of cellular multiplication that is observed as an increase in the diameter of the ovary, which is now observable as a small fruit (BBCH stage 70) [44,162]. In parallel, the volume of the embryo sac increases at the expense of the innermost layers of the perisperm, which degenerate because of the pressure exerted upon them [162]. Within the embryo sac, the ovum fuses with one of the sperm nuclei from the pollen grain, giving rise to the embryo; moreover, the other sperm nucleus fuses with the two polar nuclei in the sac, forming a triploid nucleus that, upon division, gives rise to the endosperm [135]. As the endosperm subsequently develops, it gradually consumes the perisperm, reducing it to a thin layer that covers both the endosperm and the embryo (silver skin) [132,135]. During fruit ripening (BBCH stage 88), the seed has a length of 6.5 to 19.9 mm and a width of 4.3 to 15.3 mm [163], with mean values typically between 12.64 and 13.13 mm for length, 9.12 and 9.30 mm for width, and 5.53 and 5.56 mm for thickness, with a mass between 0.2651 and 0.3096 g at a 12% moisture content [164]. Furthermore, it has an elliptical shape, is convex on one side and is flattened on the other; a longitudinal groove with a tubular appearance is observed on the flat side [44,161]. The seed represents 55% of the dry mass of the fruit [157] and is composed of three tissues: the perisperm, endosperm, and embryo [11].

Perisperm (Integument): After fertilization and up to 90 DAF, during the rapid growth phase (BBCH stage 73), the perisperm develops from the nucellus of the ovule; it grows to occupy the entire volume of the locule, driven by the intense cell division and expansion that this tissue undergoes [44,132]. Afterward, between 90 and 150 DAF, during the endosperm filling stage (BBCH stage 77), the perisperm differentiates into an outer perisperm with smaller cells facing the endocarp and an inner perisperm with larger cells facing the endosperm, and a remnant of the cell wall also exists near the endosperm, which gradually disappears as the endosperm grows [44,132]. As development progresses toward the physiological maturity of the fruit (BBCH stage 79), between 150 and 200 DAF, the remaining outer perisperm appears as a thin, green layer surrounding the endosperm [44,132]. Up to this point, the perisperm contains chloroplasts and chlorophyll [135]. During the fruit ripening phase (BBCH stage 88), the perisperm is reduced to a thin approximately 70 µm film, composed of longitudinally organized sclerenchyma cells that are likely derived from the absorbed perisperm cells and are commonly known as the “silver skin” [44,161]. It accounts for approximately 4.2% of the dry mass of the seed [158]. For the most part, “silver skin” is composed of polysaccharides, with cellulose and hemicelluloses being the most abundant; it also contains monosaccharides (glucose, xylose, galactose, mannose, and arabinose), proteins, polyphenols, and other compounds in relatively small amounts [3,165].

Endosperm: The endosperm is a living triploid (3n) tissue that originates during fertilization from the fusion of a sperm nucleus (n) from the pollen tube with the two polar nuclei (2n) in the ovary; it can be observed between 21 and 27 DAF [161]. In the pinhead stage at 30 DAF (BBCH Stage 71), the activity of various parts of the embryo sac increased, with cell division in the endosperm being particularly notable [44,135]. During the rapid growth phase, at 75 DAF (BBCH stage 73), starch grains and chloroplasts are observed inside the endosperm; after 90 DAF, the endosperm can be distinguished and separated from the embryo, as it is a milky-colored tissue with thin cell walls [44,132,135]. Beginning at 120 DAF (BBCH stage 75), the endosperm occupies the entire space previously held by the perisperm, which becomes limited to the thin film (silver skin) that covers the seed at ripening [44,135]. In the endosperm filling stage (BBCH stage 77, between 130 and 190 DAF), the endosperm cells begin to thicken because of the deposition of complex polysaccharides such as arabinogalactans and galactomannans [44,132]. In the maturation phase at approximately 230 DAF (BBCH stage 88), the endosperm contains polyhedral cells and can be divided into a hard outer endosperm, with small, polygonal, oil-rich cells, and a soft inner endosperm with rectangular cells [11,44,132]. The walls of both cell types are partially lignified and are traversed by plasmodesmata, which establish connections that enable solute exchange between cells and tissues [166]. The deposition of large amounts of polysaccharides in the cell walls makes this tissue particularly tough [167]. The endosperm contains water, proteins, caffeine, cafferine, oils, sugars, dextrin, pentosans, cellulose, caffeine-derived acids, other acids, and minor organic components [166]. This tissue constitutes the primary storage reserve for embryo growth following germination [11].

Embryo: This structure changes in shape throughout fruit development before it reaches complete differentiation. After zygote formation, between the end of the pinhead stage and the beginning of the rapid growth phase (from 40 to 70 DAF, BBCH stages 71–73), the proembryo divides and acquires a “globular” shape. This globular embryo has a diameter of 0.3 mm and a suspensor with a length of 0.08 mm [44,168]. Between 100 and 107 DAF (BBCH stage 75), the embryo develops a “heart” shape because of the development of the cotyledonary primordia at the apical part of the embryo, which, at this time, has an average length of 0.4 mm with a 0.112 mm suspensor [44,168]. Afterward, between 105 and 116 DAF (BBCH stage 75), the cotyledons become defined (0.3 to 0.8 mm in length), and the hypocotyl (1.0 to 1.6 mm in length) is observable, with the embryo acquiring a “torpedo” shape; however, these structures have not yet reached their final length [44,168]. Beginning at 120 DAF (BBCH stage 75), the embryo achieves complete morphological development and does not change in shape until fruit ripening (BBCH stage 88; 210 DAF); by 135 DAF (BBCH stage 77), the hypocotyl (2.6 to 3.0 mm long) and cotyledons (1.5 to 2.0 mm long) reach their final size, with variations of less than 9% in the total embryo length (ranging from 4.04 to 4.37 mm) [44,168]. The embryo is located at the base of the convex side of the seed and is surrounded by the endosperm, which supplies nutrients given the scarce reserves stored by the embryo itself [11,169].

### 3.4. Fruiting and Competition with Vegetative Growth

The coffee plant typically fruits only once per node on branches that are approximately 12 months old [70], and this pattern continues throughout its life cycle, with intermissions between stumping cycles, if performed. The fruiting zone on the plant moves toward the top of the main orthotropic stem and out toward the apex of the plagiotropic branches as the individual plant grows and fruits are harvested [44]. In equatorial regions, where dry and wet periods are not well defined, reproductive growth (i.e., flowers, pinhead-stage fruits, and ripe fruits on the same branch) overlaps with vegetative growth [170,171]. However, when reproductive growth is maximized, vegetative growth is reduced, and vice versa [77,172]. These climatic conditions result in multiple flowering events throughout the year [93]. Consequently, the fruits ripen asynchronously for harvest, necessitating multiple pickings [173]. In contrast, in regions farther from the equator and closer to the tropics, the synchronization between vegetative growth and fruit production is greater because distinct seasons clearly define wet and dry periods [133]. For this reason, only one or two flowering events are typically observed per year [67]. In this context, harvesting is performed when the amount of green fruit represents only 20% of the total, and therefore, ripe cherries comprise the largest percentage of the mass to be collected [174]. In both scenarios, the growth of the leaf area is usually sufficient to support the subsequent expansion of the fruits, reducing intracompetition [36,77].

In full-sun-exposed plantations, coffee plants may fail to develop a sufficient leaf area to support high fruit loads, especially after heavy flowering [133]. This process occurs because fruits are strong sinks for nutrients and carbohydrates, competing with the growth of plagiotropic branches and the formation of leaves, even from the pinhead stage (BBCH stage 71) [175]. *C. arabica* requires approximately 20 cm^2^ of leaf area to support the development of each fruit and avoid reducing vegetative growth [176]. In plants with high fruit loads (>70%), the leaf-to-fruit ratio (LFR) is lower; as a result, intense competition from fruits for the distribution of photoassimilates decreases the growth rate of branches (<0.40 mm day^−1^) [177]. Similarly, reduced stem growth, branch dieback, and extensive root death are observed [133]. An imbalance in the LFR has also been observed with increasing crop age, as the plant transitions from a low fruit load during the first fructification (16.0 cm^2^ fruit^−1^) to a high fruit load during the second fructification (8.7 cm^2^ fruit^−1^), which results in lower growth rates for branches and a reduced stem diameter [178]. This competition also affects leaves; in plants with high fruit loads (>70%), greater leaf loss from the basal position of the branch and lower leaf production at the apex have been observed once the endosperm filling stage begins (BBCH stage 77) [44,172]. Compared with those with a low load (<40%), plants with a high fruit load (>70%) have been shown to have lower leaf concentrations of nitrogen, phosphorus, potassium, and sulfur at the end of the endosperm filling stage (BBCH stage 79) [44,179]. In the fruits themselves, an increased load has been shown to reduce the seed mass [172] and size [177] but not the mass of the whole cherry fruit [179,180]. These changes, in turn, are likely linked to the high percentages of fruits with malformed endosperm (>8%) and fruit drop (>20%) when the fruit load is high (>70%) [172,179,180,181]. Fruit drop due to high loads occurs between the end of the rapid growth phase (BBCH stage 75) and the beginning of the endosperm filling stage (BBCH stage 77), between 80 and 110 DAF [44,133]. Fruits can account for up to 36% of the total plant dry mass (including roots) under an LFR of 10 to 15 cm^2^ fruit^−1^ (an indicator of plants with a moderate to high fruit load) [182]. However, fruits do not only act as sinks. Green fruits (BBCH stages 77–79) can account for 20% to 27% of the total photosynthetic surface area of plants with high fruit loads [44,62], covering up to 30% of their own daily respiration needs and contributing 12% of their total carbon requirements [44,180].

### 3.5. Fruit Set and Fruit Drop

Fruit set in *C. arabica* is evaluated at two points in time: between five and eight weeks after pollination (initial fruit set) (BBCH stage 71) and before harvest (final fruit set) (BBCH stage 88) [183]. It is defined as the proportion of flowers that successfully form fruits relative to the total number of flowers produced [184]. The initial fruit set recorded under natural pollination conditions was 75.2% at 35 DAF during the pinhead stage (BBCH stage 71) [44,87], with very similar values (between 77.0% and 86.1%) observed during the rapid growth phase (BBCH stage 75) at 90 DAF [44,85]. During the maturation phase (BBCH stage 88), the final fruit set has shown values ranging from 50% to 57% [86,185]. However, between different regions, fruit set (both initial and final) varies widely (30–90%) [183]. Fruit set is influenced by biotic factors, including the species and cultivar, branch height on the plant, the numbers of leaves and flowers, carbohydrate availability, floral disorders, and the activity of pollinating insects, and by abiotic factors, including intense rainfall, the soil nutritional status, and abrupt decreases in air temperature [133].

Premature fruit drop occurs between 90 and 120 DAF, encompassing the pinhead stage (BBCH stage 71) and the rapid growth phase (BBCH stages 73–75); it is attributed to biotic factors, including floral anomalies, low plant starch reserves, an insufficient leaf area, high fruit loads, and the incidence of pests and diseases, and it is linked to abiotic factors, such as extreme temperatures, strong winds, severe and prolonged water deficits, soil waterlogging, and deficient or imbalanced mineral nutrition [44,133,186]. Among these factors, the most relevant for commercial plantations is the imbalance in the LFR (leaf-to-fruit ratio), which is caused by a high fruit load that occurs between the end of the rapid growth phase (BBCH stage 75) and the beginning of the endosperm filling stage (BBCH stage 77) during the period of 80 to 110 DAF [44,133]. Premature fruit drop can vary between 8% and 10%, with values sometimes reaching 25.8% to 39.8% [186]. On the other hand, fruit drop after maturity can occur during the harvest of ripe fruits (BBCH stage 88), reaching values of 8.8%; however, when the fruits remain on the plant for a longer time (22 days) (BBCH stage 89), this percentage increases to 21.3% (percentages are relative to the fruit count at the beginning of the experiment) [44,187]. Nevertheless, this area requires further investigation.

### 3.6. Factors Affecting Fruit Growth and Development

#### 3.6.1. Water Availability

In plants subjected to a continuous water deficit (<10 mm) between 57 and 80 DAF, immature fruits (BBCH stage 73) in the first part of the rapid growth phase can undergo drying and blackening that begins at the peduncle and spreads to the branch, sometimes even causing fruit abscission and leading to losses of 10% to 40% of the total fruit after 80 DAF [44,188]. A lack of water during the second part of the rapid growth phase (between 91 and 119 DAF) (BBCH stage 75) produces the disorder known as “black bean”, which affects 11.41% of the fruits that reach ripening (BBCH stage 88). In these fruits, the exocarp is slightly yellowish, and they present a blackened endosperm [44,189]. Water scarcity, especially during the rapid growth phase (BBCH stages 73–75), can reduce the fresh mass and volume of the fruits, which in turn decreases the internal volume of the locules available for future endosperm development [44,176]. In this context, the fruits exhibit defects such as the so-called “empty bean” (no endosperm formation in one or both locules) and the “partially formed bean” (partial endosperm formation in one or both locules, without complete filling), as well as the formation of “small beans” (a higher frequency of fruits smaller than the standard size) [115]. In later stages of fruit development, defects due to the water deficit are less frequent and are limited to the drying of the pericarp [77]. The fruit ripening time tends to decrease in regions with a greater water deficit [190]. Therefore, rainfall during the maturation season is a key ecological factor determining the interval between flowering and fruit ripening in coffee [36]. On the other hand, excess water after dry, warm environmental conditions between the end of the endosperm filling stage (BBCH stage 79) and the ripening stage (BBCH stage 88) can cause fissures in the pericarp, a phenomenon known as “skin cracking” [191]. However, rainfall or irrigation has been shown to have less of an effect on the harvest time than nocturnal temperatures [192].

#### 3.6.2. Environmental Temperature

The most frequent effects of high air temperatures on coffee fruit include sunburn on the epidermis, problems with endosperm filling, and a reduction in ripening time [193]. Early fruit ripening occurs when the air temperature and evapotranspiration are high, when the altitude is moderate, and when a mild water deficit occurs [190]. In 15 coffee-growing localities in Colombia, the number of days between flowering (BBCH stage 60) and fruit ripening (BBCH stage 88) for the same cultivar varied between 204 and 266 DAF, a variation that was linked to the air temperature at the site [44,77]. Thus, the same cultivar will exhibit different ripening times depending on the environmental conditions of the area [190]. In shade-grown coffee plantations, incident radiation is reduced by half, and the temperature around the fruits decreases by several degrees (≈4 °C), slowing the pericarp ripening process and allowing the endosperm more time to complete its filling [194]. In cool, high-altitude regions, fruit ripening can occur two to three months later than in warm, low-altitude regions, regardless of the cultivar [190,195]. Conversely, in plantations at lower altitudes (1270–1630 m vs. 1590–1730 m) with similar water availability, higher temperatures facilitate a shorter maturation period, which reduces endosperm filling and, consequently, the seed size [196]. Temperatures above 23 °C accelerate fruit development, resulting in the production of beans with lower masses and a diminished sensory cup quality [50]. During a short maturation period, the biosynthesis of tryptophan and chlorogenic acid is inadequate, resulting in a reduced beverage quality [197]. Nocturnal temperature, as evaluated by the Warming Night Index (WNI), is strongly related to the onset of the harvest. For instance, in zones with a WNI of ~18 °C, the harvest occurred two months earlier than in zones with a WNI of ~14 °C, which affected endosperm filling and, consequently, the sensory quality of the beverage [192].

#### 3.6.3. Genotype

The time ripening varies between genotypes. For example, the cultivars Catucaí 785/15 and 24/137, with flowering (BBCH stage 60) to ripening (BBCH stage 88) cycles of 220 and 227 DAF, respectively, were classified as early-ripening; meanwhile, Palma III, Arara, and Sabiá have cycles between 238 and 254 DAF were late-ripening [44,198]. The differences between early- and late-maturing cultivars can even exceed 30 days [190]. Another way to examine the time elapsed between phenological phases is by using GDD, a measure of accumulated heat that correlates with different stages of fruit development and with the sucrose concentration in the endosperm [152]. Between flowering (BBCH stage 60) and physiological maturity (BBCH stage 79), the accumulation of thermal units differed between the cultivars, ranging from 1579 to 2634 GDD (between 123 and 217 DAF) [44,198] (10.5 °C base temperature). The maturation cycle of the composite variety Colombia was completed (BBCH stage 60–BBCH stage 88), with 2836 GDD during the main harvest [130]. For the early-ripening cultivar Mundo Novo IAC 376-4, maximum sucrose accumulation in the endosperm was reached at 2790 GDD (213 DAF), whereas for the late-maturing Obatã IAC 1669-20, it occurred at 3090 GDD (249 DAF) in Brazil, when a 10.5 °C base temperature was used; in both cases, these maxima coincided with fruit ripening (BBCH stage 88) [44,152]. The highest rate of sucrose accumulation in the endosperm occurs during the color transition from yellow-green (BBCH stage 81) to cherry red (BBCH stage 88), which is associated with the expression of enzymes that synthesize this compound [44,152].

## 4. Conclusions and Perspectives

Coffee is among the world’s principal commodities, supporting millions of smallholder producers in various countries. Consequently, knowledge of the reproductive cycle of coffee plants is crucial for guiding research focused on addressing ecophysiological challenges such as asynchronous flowering, heterogeneous fruit ripening, early ripening, deficient endosperm filling, fruit drop, and resource competition. This understanding will allow for the design of strategies for both agronomy and crop genetic improvement—with minimal impacts on coffee growth and productivity—thereby maintaining sustainability in a climate change scenario. In this context, interest in investigating how to improve the synchronization of flowering through irrigation to promote homogeneous fruit ripening at harvest time is growing; likewise, the use of hormones or other growth regulators to stimulate or delay fruit ripening should be explored. Similarly, the increase in ambient temperature and its negative effects on endosperm filling and ripening time have driven the use of phenological records to model the effects of climate change. In all these cases, the analysis of the phenological description of flowering and fruiting is crucial for determining the appropriate timing for applying treatments, defining response variables (e.g., number of flowers and number of ripe fruits), establishing evaluation times, and setting the monitoring frequency. Finally, this review serves as a foundational resource based on the described phenological knowledge of *C. arabica* flowering and fruiting to guide future research.

## Figures and Tables

**Figure 1 plants-14-03396-f001:**
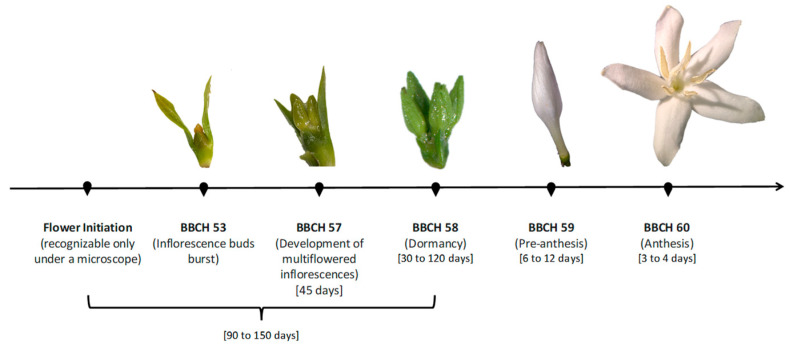
Key phenological phases of coffee flower development of the cultivar Castillo^®^ Centro in Colombia and a description of the BBCH scale associated with the stages of coffee flowers (photos courtesy of Andrés Felipe León-Burgos). The duration of a stage or set of stages appears in brackets.

**Figure 2 plants-14-03396-f002:**
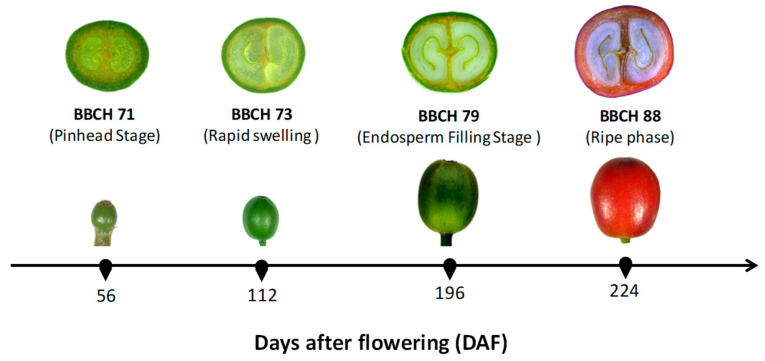
Key phenological phases of coffee fruit development of the cultivar Castillo^®^ Centro concerning days after flowering in Colombia and a description of the BBCH scale associated with the principal growth stages of coffee fruit (photos courtesy of Andrés Felipe León-Burgos).

## Data Availability

No new data were created or analyzed in this study.

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
