# Peer review of "Flowering and Fruiting of Coffea arabica L.: A Comprehensive Perspective from Phenology"

_plants, 2025, doi:10.3390/plants14213396_

Round 1

Reviewer 1 Report

Comments and Suggestions for Authors

Excellent review. The short explication about BBCH and its stages is missing. Also, figures are not used in the text and not connected to the text, to BBCH stages. Personally, I would like to have more illustrations of whole process, if possible that authors obtain some not previously published. Also, figures need temporal (and eventually seasonal) marks, as a graduate scale under the illustrations (similar as in Figure 2), where terms and BBCH numbers would be explained together.

Abstract: Why using one German phenological perspective applied for industries? When abbreviating, firstly write the full term after that the abbreviation. Write extended and scale in lowercase letters.

Introduction:

Line 30: You are talking about three species and reported two that sum 100%.

Line 36: Maybe one 0 is missing? 50 or 500 thousand years?

Line 72: Maybe include ‘belonging to’

Line 74, 84…: Arabic or Arabica coffee

Line 119: Please, a full term for BBCH

Line 140, 149, 159…: Which process? Please, do not start paragraph, or subchapter with ‘this’. Grammar, meanings…

Line 145: C. arabica is short or long day species? How 13 hours? What is the meaning of photomorphogenesis for this species?

What is the meaning of Figure 1 and Figure 2 and what are their relation to the text? They must be used in the text. Who made such illustration?

Figure 1: I recommend you to put phenology in temporal scale, adding the time of year in Colombia (and to underline that is in Colombia), and adding the duration of each phase.

Line 168: Messy part of the text: Asynchrony is or not related to floral cluster?

Line 213: Resume growth – find better term, or verb time

Line 279: More than?

Line 284: Insolation as duration in number of hours during days or sum… or intensity? Please, define.

Line 334… You do not need to abbreviate any term that is not cited anymore in the further text (ITCZ)

Line 426: stomatal density of fruits or leaves?

Line 437: Weeks after flowering?

Line 537: Here the stages are missing and temporal scale is lost. Please, include.

Try to substitute weight by mass throughout the whole manuscript, where possible.

Line 714: Here and everywhere where it is missing, please add BBCH stage

Line 800: also BBCH stage 88?

Lines 809-812: Colombia or Brazil?

Lines 836-838. Maybe you can improve this last phrase, it is not in accordance with the total work.

Details in 'pdf' file

Author Response

In the document, the responses to the reviewer 1 are highlighted in yellow.

Excellent review.

The short explication about BBCH and its stages is missing.

R/. A brief explanation of the BBCH stages has been added to the text.

Also, figures are not used in the text and not connected to the text, to BBCH stages.

R/. References to the figures have been added throughout the text.

Personally, I would like to have more illustrations of whole process, if possible that authors obtain some not previously published.

R/. Unfortunately, we do not have additional illustrations of the process.

Also, figures need temporal (and eventually seasonal) marks, as a graduate scale under the illustrations (similar as in Figure 2), where terms and BBCH numbers would be explained together.

R/. We regret not having scales for the illustrations, because the phenological scales we consulted did not provide them. We appreciate the reviewer's observation and will apply this feedback in future work. In Figure 1, we have added the duration of each stage directly to the figure.

Abstract: Why using one German phenological perspective applied for industries? When abbreviating, firstly write the full term after that the abbreviation. Write extended and scale in lowercase letters.

R/. We agree with this comment.

Introduction:

Line 30: You are talking about three species and reported two that sum 100%.

R/. It was corrected in the new paper version.

Line 72: Maybe include ‘belonging to’

R/. We agree with this comment.

Line 74, 84…: Arabic or Arabica coffee

R/. It was corrected in the new paper version.

Line 119: Please, a full term for BBCH

R/. It was corrected in the new paper version.

Line 140, 149, 159…: Which process? Please, do not start paragraph, or subchapter with ‘this’. Grammar, meanings…

R/. It was corrected in the new paper version.

Line 145: C. arabica is short or long day species? How 13 hours? What is the meaning of photomorphogenesis for this species?

R/. A paragraph was added explaining that C. arabica is a short-day species.

What is the meaning of Figure 1 and Figure 2 and what are their relation to the text? They must be used in the text.

R/. References to the figures have been added throughout the text.

Who made such illustration?

R/. It was corrected in the new paper version.

Figure 1: I recommend you to put phenology in temporal scale, adding the time of year in Colombia (and to underline that is in Colombia), and adding the duration of each phase.

R/. It was corrected in the new paper version.

Line 168: Messy part of the text: Asynchrony is or not related to floral cluster?

R/. The section of the floral cluster was deleted from the text.

Line 213: Resume growth – find better term, or verb time

R/. It was corrected in the new paper version.

Line 279: More than?

R/. It was corrected in the new paper version.

Line 284: Insolation as duration in number of hours during days or sum… or intensity? Please, define.

R/. The units were added to the text.

Line 334… You do not need to abbreviate any term that is not cited anymore in the further text (ITCZ)

R/. It was corrected in the new paper version.

Line 426: stomatal density of fruits or leaves?

R/. It was corrected in the new paper version.

Line 437: Weeks after flowering?

R/. It was corrected in the new paper version.

Line 537: Here the stages are missing and temporal scale is lost. Please, include.

R/. It was corrected in the new paper version.

Try to substitute weight by mass throughout the whole manuscript, where possible.

R/. It was corrected in the new paper version.

Line 714: Here and everywhere where it is missing, please add BBCH stage

R/. It was corrected in the new paper version.

Line 800: also BBCH stage 88?

R/. It was corrected in the new paper version.

Lines 809-812: Colombia or Brazil?

R/. It was corrected in the new paper version.

Lines 836-838. Maybe you can improve this last phrase, it is not in accordance with the total work.

R/. These lines were deleted from the text.

Reviewer 2 Report

Comments and Suggestions for Authors
  1. the manuscript would benefit from more critical analysis rather than primarily descriptive synthesis. For instance, when discussing contradictory findings in the literature regarding predawn leaf water potential thresholds for flowering, the authors list various values without sufficiently analyzing why these discrepancies exist or which conditions might explain the variation. A more analytical approach would enhance the scientific contribution beyond compilation.
  2. The section on flower development provides detailed morphological descriptions, but the manuscript could better integrate molecular and physiological mechanisms with phenological observations. For example, while the authors mention abscisic acid involvement in dormancy and ethylene in fruit ripening, these physiological processes could be more thoroughly connected to the phenological stages described. This integration would provide readers with a more mechanistic understanding of the developmental transitions.
  3. The treatment of temperature effects on phenology is generally thorough, but the manuscript would benefit from more discussion of how climate change projections specifically threaten coffee phenology. While the introduction mentions climate change as a challenge, the body of the review does not sufficiently return to this theme with specific projections or recommendations. Given that this is highlighted as motivation for the review, a more robust treatment of climate change implications throughout would strengthen the manuscript's relevance.

Author Response

In the document, the responses to the reviewer are highlighted in purple.

The manuscript would benefit from more critical analysis rather than primarily descriptive synthesis. For instance, when discussing contradictory findings in the literature regarding predawn leaf water potential thresholds for flowering, the authors list various values without sufficiently analyzing why these discrepancies exist or which conditions might explain the variation. A more analytical approach would enhance the scientific contribution beyond compilation.

R/. Thank you for the comment. We have added a paragraph elaborating on gene expression under different water potentials (lines 214–226).

The section on flower development provides detailed morphological descriptions, but the manuscript could better integrate molecular and physiological mechanisms with phenological observations. For example, while the authors mention abscisic acid involvement in dormancy and ethylene in fruit ripening, these physiological processes could be more thoroughly connected to the phenological stages described. This integration would provide readers with a more mechanistic understanding of the developmental transitions.

R/. Thank you for your thoughtful comment. In response, we have included a paragraph discussing the relationship between ABA, ACC contents, and ethylene (lines 337–349).

The treatment of temperature effects on phenology is generally thorough, but the manuscript would benefit from more discussion of how climate change projections specifically threaten coffee phenology. While the introduction mentions climate change as a challenge, the body of the review does not sufficiently return to this theme with specific projections or recommendations. Given that this is highlighted as motivation for the review, a more robust treatment of climate change implications throughout would strengthen the manuscript's relevance.

R/. We appreciate the comment. However, the objective of the present study is to focus on the crop's phenological responses at the plant and crop levels, not on the direct implications of climate change. Therefore, to avoid any confusion, we have removed the paragraph in the Introduction that referred to climate change (lines 127–132 of the original text) and the corresponding text in the Conclusions (lines 836–838 of the original text).

Round 2

Reviewer 2 Report

Comments and Suggestions for Authors

Accept in this format.